# Edge-Trimmed Nanogaps in 2D Materials for Robust, Scalable, and Tunable Lateral Tunnel Junctions

**DOI:** 10.3390/nano11040981

**Published:** 2021-04-10

**Authors:** Hai-Thai Nguyen, Yen Nguyen, Yen-Hsun Su, Ya-Ping Hsieh, Mario Hofmann

**Affiliations:** 1Department of Materials Science and Engineering, National Cheng Kung University, Tainan 70101, Taiwan; haithai6249@gmail.com (H.-T.N.); yhsu@mail.ncku.edu.tw (Y.-H.S.); 2Department of Physics, National Taiwan University, Taipei 10617, Taiwan; yenngxc@gmail.com; 3Institute of Atomic and Molecular Sciences, Academia Sinica, Taipei 10617, Taiwan; yphsieh@gate.sinica.edu.tw

**Keywords:** nanogap, tunneling devices, break junctions, 2D materials

## Abstract

Lateral tunnel junctions are fundamental building blocks for molecular electronics and novel sensors, but current fabrication approaches achieve device yields below 10%, which limits their appeal for circuit integration and large-scale application. We here demonstrate a new approach to reliably form nanometer-sized gaps between electrodes with high precision and unprecedented control. This advance in nanogap production is enabled by the unique properties of 2D materials-based contacts. The large difference in reactivity of 2D materials’ edges compared to their basal plane results in a sequential removal of atoms from the contact perimeter. The resulting trimming of exposed graphene edges in a remote hydrogen plasma proceeds at speeds of less than 1 nm per minute, permitting accurate control of the nanogap dimension through the etching process. Carrier transport measurements reveal the high quality of the nanogap, thus-produced tunnel junctions with a 97% yield rate, which represents a tenfold increase in productivity compared to previous reports. Moreover, 70% of tunnel junctions fall within a nanogap range of only 0.5 nm, representing an unprecedented uniformity in dimension. The presented edge-trimming approach enables the conformal narrowing of gaps and produces novel one-dimensional nano-trench geometries that can sustain larger tunneling currents than conventional 0D nano-junctions. Finally, the potential of our approach for future electronics was demonstrated by the realization of an atom-based memory.

## 1. Introduction

Carrier transport through atomically thin barriers proceeds by quantum mechanical tunneling, and the dependence of the tunneling current on barrier dimension and electrostatics are the foundation for scanning tunneling microscopes [1], tunneling diodes [2], Josephson junctions [3], etc. The established sensitivity on electronic properties of the barrier, the high achievable signal fidelity, and the spatial confinement afford the vision of novel electronic devices and sensors. The application of molecules as a barrier between two electrodes permits both the fundamental investigations of carrier transport and the realization of sophisticated molecular electronic devices [4,5,6,7]. Moreover, the combination of tunneling junctions with proteins or DNA has been proposed as a route for biomarker detection [8] and sequencing [9].

To realize electrodes with atomic-scale lateral gaps, different strategies have been reported. Nanogaps have been formed by focused ion beam (FIB) milling [10], feedback-controlled electro-burning [11], and e-beam lithography [12]. To date, the most commonly employed approach to producing lateral nanogaps are mechanically controlled break junctions(MBJs), where mechanical strain is employed to fracture an electrode at a pre-determined position [13].

One common challenge for all approaches is the scalable fabrication of identical devices due to the high sensitivity of nanogap performance on minute variations in patterning accuracy, pre-produced electrode dimensions, and environmental effects. This issue is exemplified in the low success rate of nanogap formation even under seemingly identical conditions. Thus far, the highest production yields of lateral tunneling junctions have been 7% [14], and significant advances have to be made to permit the successful integration of multiple nanogap devices into a complex circuit.

A potential solution to these challenges lies in the recent discovery of 2D materials [15,16,17,18]. These atomically thin materials have shown the ability to produce high-quality lateral junctions due to their highly stable and inherently sharp edges [15,16,17,18,19,20]. Moreover, the distinct difference in chemical properties between edges and the basal plane of 2D materials [19,21] permits the exclusive manipulation of edges by chemical methods [19].

We here report the fabrication of lateral nanogaps with atomic dimensions in large arrays with yield rates of 97%. The selective plasma-assisted removal of atoms from the edge of 2D materials permitted the trimming with atomic precision, enabling unprecedented control over the nanogap dimension. The conformal removal process furthermore enabled the formation of one-dimensional nanogaps that exhibit larger tunneling currents than conventional 0D constrictions. Carrier transport reveals that electrons traverse the thus formed nanogaps exclusively through direct tunneling, as evidenced by temperature-independent currents. Finally, we demonstrate the ability to produce arrays of lateral tunnel junctions with a precision of 0.5 nm over centimeter-scale and apply our advance to atomic-scale memory.

## 2. Results and Discussion

Lateral tunneling junctions were fabricated from graphene, a two-dimensional carbon allotrope. Graphene was chosen due to the large tunneling probability that originates from its small effective mass [20]. Moreover, graphene exhibits highly edge-selective reactivity to plasma and chemical etching processes [22,23] that we subsequently exploited as a novel route toward nanogap formation.

We present a novel approach to producing atomic-scale nanogaps by combining bottom-up and top-down patterning approaches. First, graphene was synthesized by chemical vapor deposition on copper foil following previous reports [24,25] and transferred onto p++-Si/300 nm SiO_2_ wafers. Edges were introduced into the graphene by lithographically patterning photoresist and removing the unprotected graphene through oxygen exposure. Subsequent deposition of metal contacts (Cr/Au, 3 nm/30 nm) using the same photoresist patterns results in the self-aligned formation of 1D edge contacts to the 2D material [26]. After lift-off of the first photoresist, a second photolithography step was employed to protect one edge contact (Figure 1a).

The thus-prepared graphene/Au edge contact was then exposed to a remote hydrogen plasma with a separation of 40 cm between the plasma source and the substrate. A continuous hydrogen flow of 5 sccm at a chamber pressure of 3 mTorr was maintained to generate a flux of ions from the plasma source to the substrate (Figure 1b). This process employs similar conditions that lead to the narrowing of graphene nanoribbons [27]. It is therefore expected that a gap between the metal contact and the graphene forms by selectively removing atoms from the initially formed edge rather than the basal plane (Figure 1c).

We observe a sharp interface between the Au contact and the graphene by scanning electron microscopy (Figure 1d) and proceed to investigate the formation of a gap between them.

Characterizing nanogaps at the interface between the contact and graphene is a significant challenge to imaging techniques. Atomic force microscopy (AFM) is an established technique to study nanostructures, but it cannot image the edge of the contact as the lateral extend of the AFM tip is larger than the nanogap. Additional challenges in the sensitivity of the amplitude feedback lead to a significant overestimation of the gap extent (Figure 2a) and cannot be used to quantify the gap dimension. Instead, we employ carrier transport measurements due to the high sensitivity of the technique to studying the size and quality of tunneling junctions [7,8,9].

After the etching process, we conduct electrical transport measurements and observe a non-linear trend of current with voltage (Figure 2b). Several conduction modes could be the origin of this trend, such as Schottky emission at incomplete nanogaps, thermally activated hopping through defects, Fowler–Nordheim (FN) type thermal emission, or tunneling through the gap [28]. To identify the dominant transport mechanism, we conduct temperature-dependent carrier transport measurements. Figure 2d demonstrates the absence of temperature dependence on carrier transport in a range from 110 to 300 K. This observation is different from the temperature-dependent resistance of graphene [29] and indicates that electron transport is dominated by direct tunneling through a sizable barrier. In this condition, the current dependence is given by Simmon’s model [30].
(1)$$I\;\propto \;V\;\exp\left[-\frac{2d\sqrt{2m\phi B}}{\hbar }\right]$$

According to Equation (1), a linear trend in the FN-plot is expected, and we indeed observe linearity when graphing ln($\frac{I}{V^{2}}$) vs. ln($\frac{1}{V}$) (Figure 2c). This result confirms the high quality of the nanogap since direct tunneling is only dominant in the absence of leakage pathways and defects. From the slope of the FN-plot, we extract a barrier height of ϕ ≈ 2.1 eV, which is in agreement with previous results [31] and a barrier width of 1 nm. Using these extracted parameters, a conventional tunneling junction will experience a current of approximately 3 nA [32,33]. Our nanogaps, however, yield almost 100 times higher currents, which suggests a change in morphology. Due to the uniform removal of carbon atoms from the edge of graphene, uniform widening of the nanogap is achieved that results in a 1D nano-trench where tunneling proceeds in parallel at a multitude of locations, as confirmed later on.

After establishing the properties of the tunneling junction, we demonstrate the fine tunability of the nanogap width through control of the etching process. Figure 3a shows the current-voltage characteristics of the same nanogap after various etching times. A decrease in current suggests that the nanogap width is increasing, resulting in a lowered tunneling probability.

We extract the gap dimensions from different samples that were etched for various times using the aforementioned approach (Figure 3b). We find that 66% of all investigated gaps show comparable dimensions after etching, with the rest exhibiting over-etching. This success rate and reproducibility of gap formation are significantly higher than previous results, and we hypothesize that the robustness is due to the uniform plasma conditions and the conformal nature of the edge-trimming process. The results thus highlight the reliability of the nanogap formation process in producing tunneling junctions with reproducible properties.

Moreover, different etching durations are shown to produce an increase in gap size, and the trend agrees with previous observations of the graphene-edge etch rate in hydrogen plasma of 0.27 nm/min [27]. A minimum etch time of 50 s was found to be required to initiate etching, which suggests that a finite exposure is required to initiate the reaction. This result highlights the feasibility of adjusting the nanogap dimension with sub-nanometer precision through simple control of the plasma exposure and the robustness of the formation process.

The uniformity of the nanogap size along the 1D nano-trench was investigated by comparing the tunneling current at a similar gap size but different nano-trench lengths (Figure 3c). We find that a twentyfold increase in nano-trench length leads to a proportional increase in tunneling current by 20×. This observation suggests that tunneling proceeds uniformly throughout the nano-trench and indicates the uniformity of nanogap size.

To demonstrate the scalability of our approach, we produced devices at wafer-scale. A total of 30 lateral tunneling devices were distributed across a 2.0 × 3.0 cm^2^ Si/SiO_2_ substrate to emulate large-scale integration. Figure 4a shows the similarity of the I–V characteristic of those 30 nanogap devices, and we only find one device that does not exhibit significant tunneling and is thus deemed an open circuit. Even for this device, forward and reverse weeps show no significant hysteresis, indicating the absence of capacitive charging.

The variability of the remaining devices is explained by differences in their tunneling gap, and we find that all of these 30 devices (97%) exhibit gaps between 0 and 3 nm. The demonstrated device yield a significant improvement over previous arrays where only seven out of 86 break junctions fell in the same gap range [14]. Moreover, we find that 23 devices (72%) exhibiting a gap between 0.5 and 1 nm (Figure 4b), which demonstrates the ability to control nanogaps with sub-nanometer precision and also provides a route to producing <1 nm gaps for direct-tunneling studies.

Our results not only enable the scalable fabrication of electrodes that are suitable for molecular electronic devices but could open up new applications. The ability to form atomic-scale gaps could, for example, increase the density of memory devices. Current high-density resistive memory devices that rely on the one-diode-one-resistor could be implemented in a single tunneling device by exploiting the non-linearity and tunable electrostatics of tunneling junctions [34]. Moreover, lateral tunneling junctions would not need to form resistive filaments, a time and power-consuming solid-state transformation, but instead, atoms could directly move into the gap. These atoms would not even have to form a conductive chain since the presence of individual atoms would modify the tunneling process sufficiently to be identified.

Figure 4c demonstrates the realization of such atomic-scale memory. When heating a nanogap to 350 K under application of a voltage, a hundredfold increase in current can be observed. This change is permanent, and characterization of the current-voltage diagram suggests that this change is caused by a decrease in barrier width (Figure 4d) by 1 nm. This finding indicates that gold atoms electro-migrate (about 10 gold atoms [35]) into the gap where they remain (inset Figure 4d). Our result demonstrates a new type of write-once-read-many (WORM) memory. Moreover, the demonstrated process could serve as a mechanism to narrow an existing tunneling gap and could be exploited in reconfigurable electronic devices or novel sensing schemes that are based on size-exclusion.

## 3. Conclusions

In conclusion, we have demonstrated a new method to reliably generate lateral tunneling junctions that exhibit atomic size nanogaps at unprecedented precision and scale. 2D material edge contacts were selectively trimmed to produce novel 1D nano-trenches whose width could be finely controlled by the etching process. Carrier transport measurements indicate the high quality of these structures and superior uniformity of their dimensions across large samples compared to traditional patterning approaches. Our results open up new applications in electronic devices, sensing, and atomic-scale information storage.

## Figures and Tables

**Figure 1 nanomaterials-11-00981-f001:**
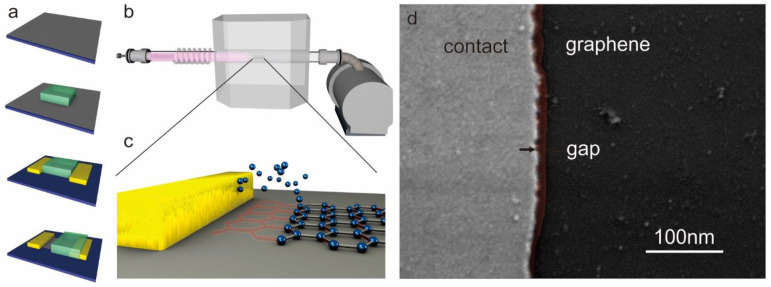
Method to producing lateral tunneling junctions. (**a**) Sequence of lithography steps to produce an edge-contacted 2D material prior to edge trimming (graphene transfer, photolithographic channel definition and isolation, edge contact deposition, lithographical protection of one contact); (**b**) depiction of the remote plasma chamber and close-up schematic of edge trimming between the metal contact and graphene edge; (**c**) schematic of the edge-trimming process that results in nanogaps between graphene and the edge contact; (**d**) scanning electron micrograph of produced structure with the indication of gap position.

**Figure 2 nanomaterials-11-00981-f002:**
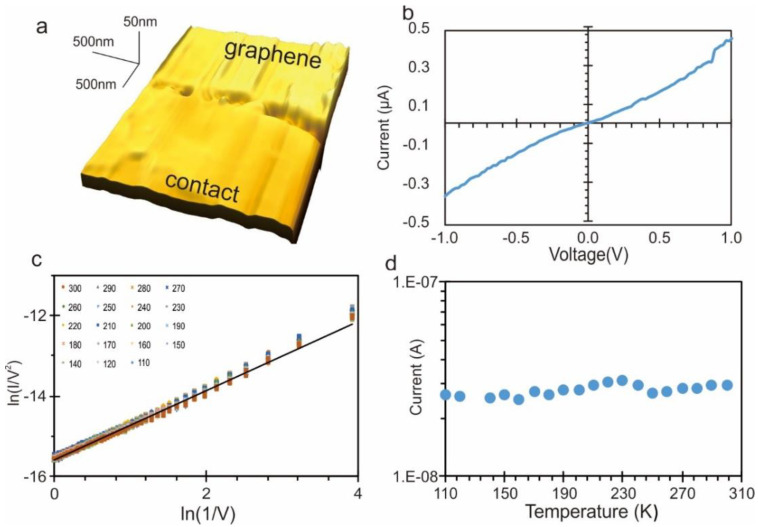
Carrier transport measurement of representative nanogap after 65s etching. (**a**) Atomic force micrograph of graphene/contact interface; (**b**) representative IV at room temperature; (**c**) FN-plot of IV at different temperatures; (**d**) current at a fixed voltage (0.2 V) vs. temperature.

**Figure 3 nanomaterials-11-00981-f003:**
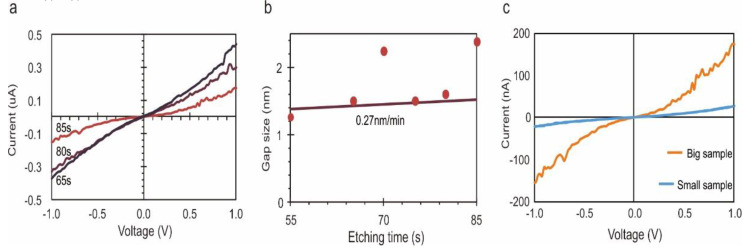
Characterization of the etching process. (**a**) IV for the same tunnel junction at different etching times; (**b**) extracted tunneling gap size from (a) as a function of etching time with overlay of previously observed edge-trimming rate Xie et al. [27]; (**c**) comparison of current-voltage characteristics for two different lateral gap widths but the same gap size.

**Figure 4 nanomaterials-11-00981-f004:**
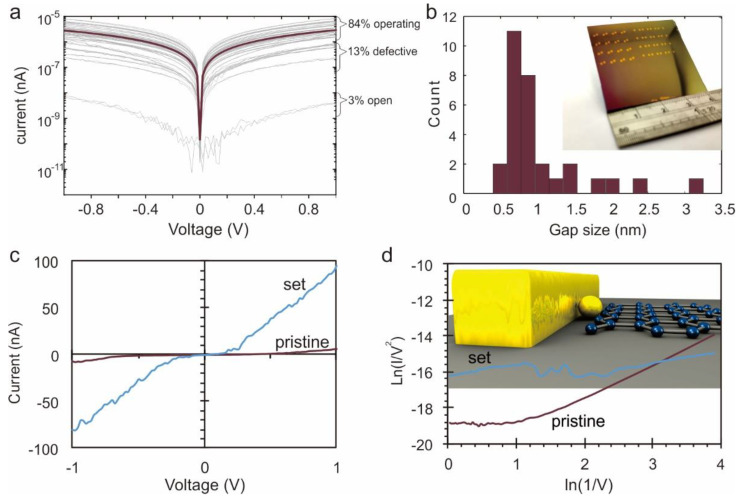
Scalability of nanogap formation approach and applications. (**a**) Overlay of 30 IVs obtained from one sample with the indication of status; (**b**) histogram of gap sizes throughout the sample indicating that 70% of devices exhibit gaps within a 0.5 nm range, (inset) photograph of sample; (**c**) realization of atomic-scale memory showing distinct IV before and after writing step, (inset) tunneling mechanism in pristine and set cases; (**d**) FN plots of pristine and set memory states, (inset) depiction of set state.

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
