# Peer review of "Edge-Trimmed Nanogaps in 2D Materials for Robust, Scalable, and Tunable Lateral Tunnel Junctions"

_nanomaterials, 2021, doi:10.3390/nano11040981_

Round 1

Reviewer 1 Report

Introduction:

  • typo in line 37 “device s[4-7]”
  • be consistent with the format. Full stop before or after [13] (line 45)?
  • Line 53 “recent discovery” – references 12-21 are mainly from 2012-2016. I would say they are slightly out of date, please update the relevant references to more recent 5 years if possible.
  • reasons for choosing graphene as 2D material and not others? A quick summary of why graphene is chosen would be great. Or maybe move line 69-71 to the introduction part.
  • Missing full stop sign after [22] in line 57.

Results and discussion.

  • more experimental details should be included to demonstrate reproducibility at other labs e.g. plasmon setting conditions, and etching conditions.
  • label the objects in Fig. 1 (a)-(b) clearly, either on the figure or in the caption for readers to follow.
  • be consistent with the format. Want space before the units or not? E.g. 40cm (line 82), 3 mTorr (line 83)
  • According to Fig. 1(d), the gap sizes are different in 400 nm length. Does the variation of gap size affect the electrical transport measurement?
  • Can gap size be uniform across a bigger size of area?
  • In Fig. 2(b), the I-V curves are asymmetric at ~0.9V, any explanation?
  • be consistent with the format. Full stop before or after [ref] (line 114, 119, 121, 145)? Please check the referencing format throughout the whole manuscript.
  • What about shorter etching time (< 65 s) in relation to tunneling probability?
  • Do the authors use the same plasma parameter as ref 27? Seems like the authors tried different etching times (shown in Fig. 3(a)), it would be better to show their actual experiment data (gap size vs etching time) as Fig. 3(b), instead of using data from Xie et al.
  • Line 149-151 mentioned 30 devices, somehow line 160 mentioned 31 devices were tested?
  • Line 160, is 31 devices right? There is 1 device that exists as an open circuit, so there should be 29 devices I suppose.
  • It would be great to mention the ideal gap size range for applications. If aiming gaps < 5 nm, then your method is like ~97% success rate. Need more introduction on what kind of parameters are you aiming for. Smaller the gap the better the performance?
  • Check sentence in line 162-164. “Moreover……” sounds like that 0.5-1 nm is undesired.
  • Most importantly, the success rate is impressive, but what the author did differently to others (e.g. Xie et al.) to achieve the success rate? Any differences in synthesis conditions (e.g. plasmon or etching) when compare with Xie et al?  Highlights should be around the synthesis parameters the authors tried to achieved such a high rate, but that didn't get clearly mention and explain. The success rate of gap formation was only explained from uniform plasma conditions and edge trimming process, which should be the fundamental processes for other labs. Why and how the authors do differently to achieve a high success rate?

Reviewer 2 Report

This paper studies a method for creating nm-size gap between graphene and metal electrode, for high-yield fabrication of tunnel junctions. The paper is clearly written and the results are promising.

I have one question/suggestion: it would be very helpful if the authors can discuss the dependence of the tunneling current on the lateral width of the junctions. If tunneling is uniform (presumably preferred), then the tunneling conductance should scale with the width. On the other hand, if tunneling happens at a single or a few points, then there'll be no such width dependence.  The authors should also discuss the significance of uniformity of tunneling in real-life applications.

Reviewer 3 Report

The authors presented a controllable method to fabricate nanometer-sized gaps between electrodes and graphene. The carrier transport characterizations are adopted to prove the tunneling junction. I think this paper can be improved after solving the following questions:

  1. The authors should provide the temperature-dependent electrical transport results of graphene without the gap. The authors need to compare the results with and without the gap, then it will demonstrate the formation of a tunneling junction. 
  2. There are several typos in the manuscript. In the abstract: "Carrier transport measurements reveal the high quality of the,"
